# Sensor Placement in an Irregular 3D Surface for Improving Localization Accuracy Using a Multi-Objective Memetic Algorithm

**DOI:** 10.3390/s23146316

**Published:** 2023-07-11

**Authors:** Paula A. Graça, José C. Alves, Bruno M. Ferreira

**Affiliations:** 1INESC TEC—Institute for Systems and Computer Engineering, Technology and Science, Rua Dr. Roberto Frias, 4200-465 Porto, Portugal; bruno.m.ferreira@inesctec.pt; 2Faculty of Engineering, University of Porto, Rua Dr. Roberto Frias, 4200-465 Porto, Portugal; jca@fe.up.pt

**Keywords:** 3D sensor placement, underwater acoustic localization, multi-objective optimization, memetic algorithm, ultra-short baseline, Fisher information matrix, Cramer-Rao Lower Bound, incidence angle

## Abstract

Accurate localization is a critical task in underwater navigation. Typical localization methods use a set of acoustic sensors and beacons to estimate relative position, whose geometric configuration has a significant impact on the localization accuracy. Although there is much effort in the literature to define optimal 2D or 3D sensor placement, the optimal sensor placement in irregular and constrained 3D surfaces, such as autonomous underwater vehicles (AUVs) or other structures, is not exploited for improving localization. Additionally, most applications using AUVs employ commercial acoustic modems or compact arrays, therefore the optimization of the placement of spatially independent sensors is not a considered issue. This article tackles acoustic sensor placement optimization in irregular and constrained 3D surfaces, for inverted ultra-short baseline (USBL) approaches, to improve localization accuracy. The implemented multi-objective memetic algorithm combines an evaluation of the geometric sensor’s configuration, using the Cramer-Rao Lower Bound (CRLB), with the incidence angle of the received signal. A case study is presented over a simulated homing and docking scenario to demonstrate the proposed optimization algorithm.

## 1. Introduction

Underwater localization is a challenging task due to the varying environment characteristics and technological restrictions. As electromagnetic signals are highly absorbed underwater, a global navigation system is not available and localization techniques are commonly chosen depending on the expected range of operation, to improve the estimation performance. Typical short range localization approaches, up to a few meters, can use optical or radio frequency signals, which are not suitable for transmission over long distances [1]. While optical wave propagation is highly dependent on the water turbidity and light conditions, with a typically effective transmission range to some tens of meters, radio frequency signals are very attenuated, especially in salty water, and can also be affected by surrounding electromagnetic fields [2,3]. Therefore, most localization techniques rely on the transmission of acoustic signals, which are capable of traveling distances of a few kilometers and are insensible to water visibility conditions. However, these are also affected by the underwater acoustic channel, causing time-varying and frequency distortion, multipath loss and fading [4]. These phenomena cause interference in the acoustic signals that affects the integrity of the signal and compromises detectability.

Due to the varying environment conditions and implementation limitations, many localization applications rely on range-based methods [5]. The most popular approaches resort to acoustic transmitters and receivers to estimate time delay, such as time of arrival (TOA) and time difference of arrival (TDOA), and compute range, relative position or localization based on these measurements. Positioning approaches are usually categorized according to the baseline distance between sensors, and can be divided into long baseline (LBL), short baseline (SBL) and ultra-short baseline (USBL). The latter implies typical baselines of tens of centimeters and is commonly used when deploying sensors in constrained surfaces or structures, such as autonomous underwater vehicles (AUVs). Although USBL is more prone to uncertainty introduced by environment measurements and installation errors, it requires a lower setup effort and, by employing compensation mechanisms and calibration routines, it is possible to considerably improve its localization performance [6].

The relative geometric position between sensors is a critical aspect to minimize localization error in positioning and localization approaches. One of the most commonly used metrics to evaluate localization uncertainty of sensor configurations is the Cramer-Rao Lower Bound (CRLB) [7], defined as the inverse Fisher information matrix (FIM). For a specific sensor configuration, the CRLB characterizes the spatial variance distribution of the estimates, for any efficient and unbiased estimator, by quantifying the information that the configuration can perceive about an unknown variable. This technique has been used for underwater sensor positioning optimization [8,9,10,11,12,13], usually in LBL approaches. However while optimality of sensor placement in a 2D surface or 3D volume is commonly addressed in the literature, the considered distances between sensors are incompatible with the deployment in 3D constrained bodies, such as AUVs or other structures, which show challenges that hamper the creation of benchmarks for optimality. Firstly, as underwater vehicles can have a complex morphology, such as asymmetric or bio-mimetic shapes [14], sensor placement in these structures is restrained and there is a high chance of occlusion between the receiving sensors and an acoustic emitter [15]. Additionally, underwater applications usually use compact arrays and acoustic modems [16,17,18], which limit the customisation of relative geometry between sensors. Although there are studies on the optimal geometric relation between individual receiving sensors of a constrained array [19,20], the optimal placement of spatially independent sensors in an AUV or structure, to improve the localization performance, is not a considered issue in literature to the best of the authors’ knowledge. We argue that the work presented herein is the first to propose a methodology for optimized receiver placement on an irregular 3D surface of a vehicle.

Several optimization and search methods have been exploited throughout the years for sensor placement, as previously mentioned, including single and multi-objective optimization as well as non-traditional algorithms. Since this problem typically has a solution domain with a provable NP-hard property [21,22,23], as it scales with the number of parameters and admissible area, heuristic methods have been exploited to solve it. In the present work, it was observed that an exhaustive optimization search would take around 10 h to find the global optimal solution for a very simple data set of 100 points, corresponding to approximately 4 million possible solutions, with a fully dedicated Intel Core I5-11600. However, the expected data sets are usually thousands of times larger making it not a suitable strategy for this work. When exhaustive search is not viable, heuristic search methods may be a trade-off for reaching good, although sub-optimal, solutions in acceptable running times. Therefore, we will focus on evolutionary algorithms, a group of heuristic search methods inspired by Darwin’s natural selection [24]. Typically, these algorithms consider a population of solutions that are propagated and improved throughout generations, leading to the survival of the fittest ones, according to an optimization function. Evolutionary algorithms have been applied in the field of underwater sensor placement as a way to optimize solutions and improve computational performance, among others. Examples are particle swarm optimization (PSO) algorithms, proposed in [25,26], that select the optimal value from the swarm as the goal for the next generation and are usually suitable for continuous domains. Alternatively, genetic algorithms as presented in [9,21,23], which are suited to discrete domains, implementing evolutionary processes inspired by biological species, that ensure a level of randomness in the process and variability in the population. Memetic algorithms (MAs) [27] are a variation of genetic algorithms, that combine evolutionary processes to refine the search and optimization. MAs employ the general procedures of genetic algorithms, allowing solution handling parallelization and diversification, that are interleaved with local search (LS) strategies, which allow us to inject intensification search phases and increase randomness in the process. This hybrid method has proven to improve the results of pure genetic algorithms [11,22].

This paper tackles the optimization of the placement of a set of underwater acoustic sensors along an irregular and constrained 3D surface, to improve the localization of an acoustic beacon. The multi-objective optimization aims to determine the sensor configuration that minimizes, or bounds, localization uncertainty, depending on the range and number of reference relative positions, while aiming for a reduced incidence angle of acoustic signals received at each sensor. This method is built on previous work [28] that exploits optimization of sensor placement for a single reference position. In order to illustrate the relevancy of this method, a case study is presented for a homing and docking scenario of an AUV, with an installed set of acoustic sensors following an inverted USBL approach. The main contributions of the present work are:(a)A novel methodology for the sensor placement problem in an irregular and constrained 3D surface, that aims to improve localization by combining the quality of the measurements, dependent on the incidence angle of acoustic signals at the receivers, and the information obtained from the measurements, given by the localization uncertainty achieved by each sensor configuration;(b)A multi-objective memetic optimization algorithm that selects the sensor configuration to be placed in an irregular and constrained 3D surface, that reduces incidence angle while: (1) Minimizing localization uncertainty for a single reference beacon position; (2) combining uncertainty minimization and bounding, based on the range of the reference beacon position;(c)The application of the proposed algorithm to select optimized configurations for an AUV, following an inverted USBL approach, in homing and docking scenarios.

Additionally, in order to achieve the main contributions of this work, a framework was developed to formulate 3D surfaces and map the incidence angles of an acoustic signal received from a fixed beacon.

The present paper is organized as follows: Section 2 defines the problem, details the developed framework for generating irregular 3D surfaces and describes the methods used to evaluate single sensors and sensor configurations. Section 3 describes the implemented memetic algorithm. Section 4 presents a case study supported by simulations and detailed the results. Lastly, Section 5 states the main conclusions and future work.

## 2. Problem Formulation

The present problem addresses the placement of acoustic sensors on an irregular and constrained 3D surface, referred to as the deployment body, to improve relative localization of an acoustic beacon. It is assumed that an approximated receiver-emitter topology is known for the intended application, meaning that the general expected orientation between the deployment body and the beacon is known. For example, during AUV deep surveys, typical operations use a support vessel acting as an anchor reference point, where the relative position can be assumed to be approximately constant. Another example is a docking operation, during which an AUV has to face the structure to be able to dock. This method is intended to evaluate the sensor configuration previously to the sensor placement, and it is useful, for instance, to install sensors in AUVs following an inverted USBL configuration, to improve the relative localization of an acoustic beacon in planned missions.

Acoustic sensors, particularly hydrophones, are typically characterized by a beam pattern that can either be omnidirectional, which allows signals to be received with any incidence angle with approximately the same attenuation, or direction, which has a varying attenuation based on the incidence angle of a received signal. In this method, since the sensors are placed on the non-planar surface of a 3D body, or even partially buried in the structure wall, it is considered that only approximately 180° is accessible for signal reception. Additionally, the surface adjacent to the sensors adds interference to the received signals, due to reflection, scattering and vibration. Therefore, as an effort to reduce interference, a directional beam pattern was conceptualized to translate the sensors’ sensitivity response as a function of the incidence angle. However, it is difficult to generalize the accurate sensor response as a function of the incidence angle and it depends on the specific sensor. Thus, we consider that the incidence angle may be translated into a measurement error of the TOA of signals arriving at the acoustic sensors. Since the TOA measurements are used to compute relative distances, the introduced errors negatively affect the relative localization between the deployment body and the acoustic beacon. In order to quantify these errors, a penalization system was created based on the attenuation that the sensor is expected to experience for varying incidence angle. This penalization will be used in the optimization function and allows to evaluate each individual candidate sensor position on the 3D deployment body based on the incidence angle, without loss of generalization as the optimization process can be applied with alternative and more complex sensor sensitivity functions.

Regarding the multi-objective optimization, two main criteria are used:Single sensor evaluation: as acoustic signals are used to estimate localization, it is considered that the quality of the measurements depends on the integrity of the signal arriving at each receiver. Since the acoustic sensors are placed in a surface, the incidence angle of arriving acoustic signals influences the quality of the measurements. Therefore, in this work this is quantified by the incidence angle between a known acoustic beacon position, that transmits acoustic signals, and each sensor placed in the 3D surface [29]. Sensor positions that lead to lower incidence angles are favoured;Sensor configuration evaluation: it is assumed that the information extracted from the measurements is quantified by the localization uncertainty obtained from each configuration of sensors. Configurations that lead to lower localization uncertainty are favoured.

It should be noted that this method does not intend to tackle common underwater acoustic channel phenomena that affects the acoustic signal propagation, such as multipath originated from reflection on obstacles in the environment, Doppler effect, fading or variable sound speed. The single sensor evaluation only contemplates interference added to the acoustic signals arriving at each sensor by multipath and reverberation phenomena originated from the physical characteristics of the surface where it is located.

In order to apply the proposed methodology, it is necessary to formulate a 3D model of the deployment body, which can be a vehicle or structure. The 3D model must be representable by discrete positions that serve as candidate positions for sensor deployment during the optimization stage, and must allow deduction of the slope of the surface; thus, the directivity of the candidate positions in relation to a reference position, in order to obtain incidence angles.

Depending on the application or mission, the acceptable localization uncertainty provided by the sensor configuration may vary, therefore absolute optimization may not be required and different constraints can be applied depending on the range between the deployment body and the beacon. In short range localization, in the range of a few meters, it can be critical to achieve minimum uncertainty of the relative position, as in scenarios of docking or navigation in a structured or confined environment [30]. Conversely, in medium/long range navigation, it can be reasonable to only seek a bounded localization error, which is enough to navigate towards a target, for example in a search, black box recovery or homing missions [31]. In applications that include both short to long range scenarios, sensor placement can be optimized for the expected relative positions along the mission. By adopting an optimization function that combines different constraints based on range, it is possible to achieve a higher adaptation of the sensor deployment to the localization requirements during a mission.

Therefore, assuming that the deployment body may have the ability to move and a single beacon is installed in the environment, several relative positions between the beacon and the deployment body can be considered for the sensor placement optimization problem. Considering a reference frame fixed in the deployment body, the relative beacon positions are represented as $\eta_{b}={\left[x_{b}\;y_{b}\;z_{b}\right]}^{T}$ and, based on the intended application, the optimization algorithm falls into one of the following approaches:Single improvement: suitable for scenarios in which it is relevant to select a sensor configuration that improves localization for a single fixed relative position between the deployment body and a beacon. Therefore, the goal is to select a sensor configuration that minimizes localization uncertainty for a single reference beacon position $\eta_{b}$, disregarding the range.Figure 1 illustrates a single improvement scenario, in which it is relevant to minimize localization uncertainty for a single relative position between the beacon and the deployment body A. Therefore, the selected sensor configuration minimizes localization uncertainty, represented by the narrow green region around the beacon, for the selected relative position. This approach is relevant for applications such as cooperative navigation, target tracking and docking, where it can be essential to improve localization for an approximately constant relative position between an acoustic transmitter and receiver.Double improvement: assumes a scenario in which it is relevant to select a sensor configuration that simultaneously improves localization for two fixed relative positions between the deployment body and a beacon, covering both short and long range. In this case, the objective is to minimize localization uncertainty for a short range position while bounding uncertainty for a medium/long range position.Figure 2 illustrates a double improvement scenario, in which a deployment body B follows a mission in proximity to a beacon. The mission requires low localization uncertainty for a short range position, represented by the green body, and has more relaxed localization needs for a medium/long range position, represented by the orange body. Therefore, the selected sensor configuration minimizes localization uncertainty for the short range position, represented by the narrow green region around the beacon, while bounding uncertainty for the long range position, represented by the larger orange region around the beacon. This scenario can be observed in applications such as homing and docking, and search and recovery missions, where distinct localization requirements may be expected depending on the relative range between an acoustic transmitter and receiver.

### 2.1. Surface Model

The 3D deployment surface represents the structure or vehicle where the sensors will be placed, which may have diverse shape complexity. Therefore, a framework was developed to create generalized representations of surfaces as 3D point clouds with defined granularity, composed by basic discretized shapes (e.g., parabolic volumes, cones, cylinders, etc.), preserving the relevant features of the real structures or vehicles. It also introduces the possibility to define forbidden areas for sensor placement, avoiding placing sensors near noise sources, such as thrusters, or in fragile zones, such as wings, and allows areas to be indicated where other sensors are already placed, such as cameras or sonar.

The volumetric 3D body, grounded on the xy plane on a body-fixed reference frame, is formed by revolution around the positive *z* axis of a discrete 2D function defined in the positive space of the yz plane. Shapes that cannot be generated by revolution are also admissible in the context of this work; however, with a more complex creation process.

Assuming that the acoustic sensors are placed on the surface of the 3D body, it is important to know the spatial orientation of the locations on the body surface where the sensors can be placed. Since the 3D point cloud does not provide direct information regarding the slope of the surface along the body, it is necessary to generate a surface that allows incidence angles of the acoustic wave reaching the deployment body to be obtained. Therefore, a continuous surface is generated from the 3D point cloud by applying Delaunay triangulation [32]. This allows to create non-overlapping triangles between each group of three points, that is followed by constraining the triangulation for non-convex domains. Each triangle can be interpreted as a plane with a centroid and a normal vector $\vec{n}$, allowing information regarding the incidence angle of the acoustic wave reaching the 3D body to be deduced. The resulting continuous surface has a level of detail determined by the number of generated triangles and the triangle centroids represent all possible positions for sensor placement, as illustrated in Figure 3. These are defined as $\eta_{p_{k}}={\left[x_{p_{k}}\;y_{p_{k}}\;z_{p_{k}}\right]}^{T}$, with *k* = 1, …, *K*, where *K* is the number of points in the point cloud P. For each candidate configuration, composed by *S* sensors, the positions of the sensors are expressed as $\eta_{s_{i}}={\left[x_{s_{i}}\;y_{s_{i}}\;z_{s_{i}}\right]}^{T}$, where *i* = 1, …, *S* and $\eta_{s_{i}}\in P$. Throughout this paper, the acoustic sensors are represented by points and considerations about the dimension of the sensors will be taken into account further on.

### 2.2. Single Sensor Evaluation

When evaluating a sensor configuration, it is relevant to consider the relative orientation between the acoustic sensors and the transmission beacon. We assume that the sensors are deployed on the surface of the deployment body and are parallel to the surface, as illustrated in Figure 4. Therefore, we consider that the individual sensors have a sensitivity response that is a function of the incidence angle, as it is relevant that each sensor receives the acoustic signal with sufficiently low added noise and interference to allow extracting accurate information. In this regard, the proposed method uses the incidence angle of a signal emitted from a known acoustic beacon position to quantify the quality of the received signal in each candidate position of the deployment body [29].

Given the point cloud P that describes the deployment surface, the incidence angle is computed for all possible sensor positions $\eta_{p_{k}}$, considering a reference beacon position $\eta_{b}$. Defining ${\vec{v}}_{bp_{k}}$ as the unit vector that connects the beacon position $\eta_{b}$ and point $\eta_{p_{k}}\in P$, it can expressed as
(1)$${\vec{v}}_{bp_{k}}=\frac{\eta_{b}-\eta_{p_{k}}}{\left\|\eta_{b}-\eta_{p_{k}}\right\|}.$$

Recalling that each triangle can be interpreted as a plane with a normal vector $\vec{n}$, the incidence angle is defined as the angle between ${\vec{v}}_{bp_{k}}$ and $\vec{n}$, placed in a triangle centroid $\eta_{p_{k}}$, as illustrated in Figure 5. The incidence angle for each $\eta_{p_{k}}$ is denoted as $\phi_{k}$ and it is calculated as follows
(2)$$\tan\left(\phi_{k}\right)=\frac{\left\|\vec{n}\times {\vec{v}}_{bp_{k}}\right\|}{\vec{n}\;\cdot \;{\vec{v}}_{bp_{k}}}.$$

In this formulation, the ideal incidence angle of a signal coming from the acoustic beacon to the centroid of the triangle is equal to 0°, hence coincident with the orientation of $\vec{n}$.

As previously mentioned, an acoustic beam pattern was formulated to describe the sensitivity response of the acoustic sensors as a function of the incidence angle of the acoustic wave arriving at the deployment body. The designed beam pattern is symmetrical with respect to 0° and has an attenuation, corresponding to the penalization level, $\rho_{k}$, that does not vary linearly with the incidence angle. The used model for the beam pattern is described by a sigmoid function (Equation (3)), with an inflection point $\phi_{ip}$ and hill slope *h* as illustrated in Figure 6. These parameters should be adapted to meet the actual sensor beam pattern.
(3)$$\rho_{k}\left(\phi_{k}\right)=\frac{100}{1+e^{\;h(\phi_{ip}-\phi_{k})}}$$

The penalization function was used to evaluate each sensor position individually. To each $\eta_{p_{k}}$ is attributed a penalization level, $\rho_{k}\in$ [0, 100]. $\rho_{k}$ weights on the objective function of the optimization algorithm and it penalizes candidate positions with $\phi_{k}$ that are more deviated from the ideal incidence angle and more prone to interference, such as positions without line-of-sight (LOS) to the beacon or too close to the surface around.

Figure 7 illustrates three beam patterns in polar coordinates, where the angle represents the incidence angle and the inverse of the radius corresponds to the penalization $\rho_{k}$. B1 is a generic omnidirectional beam pattern, B2 is the expected pattern for omnidirectional sensors that are placed on a surface and B3 represents the conceptualized beam pattern for the sensors in this work. In this case, the inflexion point in B3 is set to $\phi_{ip}$ = 45°, as we intend to greatly penalize incidence angles in the range [45, 90]°. However, it should be noted that this parameter is reconfigurable and depends on the adopted sensor model, therefore it does not compromise the generalization of the proposed method.

### 2.3. Sensor Configuration Evaluation

The geometric relationship between sensors in a configuration have a direct impact on the localization uncertainty. Therefore, the 3D sensor configuration is evaluated using the classic CRLB method, following a TOA based localization approach [7].

Assuming a time of emission $t_{0}$, the relative positions between sensor *i* and the beacon are given by ${\hat{\eta }}_{i}=\eta_{s_{i}}-\eta_{b}$, and the TOA to each sensor *i* is given as
(4)$$t_{i}=t_{0}+\frac{\sqrt{{{\hat{\eta }}_{i}}^{T}{\hat{\eta }}_{i}}}{c_{s}},$$
where the underwater sound speed, $c_{s}$, is assumed to be constant and equal to 1500 ms^−1^.

Assuming Gaussian measurement errors [13], the FIM $I\left(\eta_{b}\right)\in R^{3x3}$ is defined as
(5)$$I\left(\eta_{b}\right)=J{\left(\eta_{b}\right)}^{T}\Sigma^{-1}J\left(\eta_{b}\right),$$
where $\Sigma =diag(\sigma_{i}^{2})$ is a covariance matrix with constant values of variance, $\sigma_{i}^{2}$. $J(\eta_{b})$ is the Jacobian of the measurement vector with respect to $\eta_{b}$ defined as
(6)$$J\left(\eta_{b}\right)=\frac{1}{c_{s}}{\left[\begin{array}{c} \frac{{{\hat{\eta }}_{1}}^{T}}{\|{\hat{\eta }}_{1}{\|}^{2}}\;\ldots \;\frac{{{\hat{\eta }}_{i}}^{T}}{\|{\hat{\eta }}_{i}{\|}^{2}} \end{array}\right]}^{T}.$$

With some algebraic manipulation, $I(\eta_{b})$ can be expressed as
(7)$$I\left(\eta_{b}\right)=\frac{1}{c_{s}^{2}}\left[\begin{array}{c} \sum_{i=1}^{S}\frac{{\hat{\eta }}_{i}{{\hat{\eta }}_{i}}^{T}}{\sigma_{i}^{2}\|{\hat{\eta }}_{i}{\|}^{2}} \end{array}\right].$$

The FIM provides the amount of information that a sensor configuration can provide about the acoustic beacon position $\eta_{b}$. Therefore, we are interested in quantifying the lower bound localization uncertainty that a configuration is capable of obtaining. This is given by the CRLB, that allows to analyse the spatial variance distribution of a configuration characterized by an uncertainty ellipsoid $C(\eta_{b})$, assuming an efficient and unbiased estimator [7]. By defining the uncertainty ellipsoid as
(8)$$C\left(\eta_{b}\right)=I{\left(\eta_{b}\right)}^{-1}\;$$
and referring to its eigenvalues as $\lambda_{l}$ for l=1,…,3, then the length of the *l*-th axis of the ellipsoid is represented by $\sqrt{\lambda_{l}}$ and its direction is given by the corresponding eigenvector.

Based on these definitions, several optimality criteria are commonly contemplated to evaluate sensor configuration in localization problems [10,12,33,34]. In this work, the E-optimality criterion was selected as it evaluates quantitatively each sensor configuration by the largest uncertainty ellipsoid axis, Λ, that it generates for a defined $\eta_{b}$, denoted as
(9)$$\Lambda =\underset{l}{\arg\max}\sqrt{\lambda_{l}}.$$

These were computed using the power iteration method for computational purposes [35]. The sensor configuration optimization will then aim to select the configuration that minimizes Λ, further explained in Section 3.1.

## 3. Multi-Objective Memetic Algorithm

The addressed problem in this paper is the placement optimization of a set of acoustic receivers in an irregular and constrained 3D surface, according to the position of a beacon installed in the environment. This problem has a solution space with provable NP-hard property [21,22], as it escalates with the number of sensors, surface area complexity and granularity of the discretized surface. Since exhaustive search strategies are not suitable for these problems, heuristic methods are commonly employed since they are capable of achieving optimized solutions in a timely manner. Memetic algorithms (MAs) are a variation of genetic algorithms that add periodic stages of local search (LS) to improve the quality of the solutions and speedup the convergence process [27]. Although the general process performed is simple, they allow a high level of customization regarding the formulation of the objective function, representation of solutions and all the processes involved in the evolution and selection of new solutions.

The proposed MA presents a solution space composed by all candidate sensor configurations, that are formed by all combinations of *S* sensor positions $\eta_{p_{k}}\in P$. The final solution is given as a vector containing the *S* sensors positions $\eta_{s_{i}}$ that form the optimized configuration. The algorithm receives as inputs the coordinates of the deployment surface point cloud P, the corresponding penalization levels $\rho_{k}$ for each $\eta_{p_{k}}$ and one or two beacon position $\eta_{b}$, depending on the selected single improvement or double improvement strategies.

Adopting genetic algorithm’s notation, the algorithm iterates throughout *G* generations evolving a population of solutions, denoted *P*. *P* is composed by *N* individuals that constitute feasible solutions of the problem. Each individual represents a candidate sensor configuration that is composed by the set of positions of *S* sensors, referred to as genes. Each gene represents a candidate position for placing an independent sensor, and has an associated $\eta_{p_{k}}$ (coordinates of the sensor position) and corresponding $\rho_{k}$ (penalization level). Based on the single sensor evaluation and sensor configuration evaluation previously presented in Section 2.2 and Section 2.3, a fitness value is attributed to each individual that quantifies how optimized a configuration is for the proposed objective. The multi-objective fitness function is further detailed in Section 3.1.

Algorithm 1 presents the proposed MA. The population is initially created by forming *N* individuals, each composed of *S* genes. Instead of simply generating a random initial population, the individuals of the initial population are selected in order to ensure initial high spatial coverage of the deployment surface, by constraining the spatial location of the genes that form each individual in spaced out azimuth angles. Then during each generation, the following steps are executed:

Selection and CrossoverFor a number of iterations corresponding to *c*% of the population, two individuals are selected and crossover is performed between the individuals. The random selection method starts by sorting the individuals by descending order of their fitness value and partitioning the population into a fittest group of χ% of the individuals and another group with the remaining. One individual is randomly selected from each group, with equal probability, constituting the parents for the single-point crossover operation [36]. The crossover consists in recombining half of the genes from each parent to generate an offspring. The offspring is lastly introduced in the population by replacing the least fit individual of the current population. By creating new offspring that have intrinsically favourable and not so favourable genes, it is intended to discover new combinations of genes that lead to an improved fitness.MutationFor a number of iterations corresponding to *m*% of the population, an individual is randomly selected and mutated. The mutation consists of switching one of its genes with another randomly selected from the solution space, introducing higher gene variability to the population. The new mutated individual is introduced in the population by replacing the least fit in the population.Local SearchIn each δ generations, the *l*% fittest individuals of the population are subjected to LS. This step allows intensification of the search for an optimized solution in individuals that have the most suitable fitness. The LS algorithm is detailed in Section 3.2.

**Algorithm 1:** Memetic Algorithm
**Result:** Global optimized sensor configuration


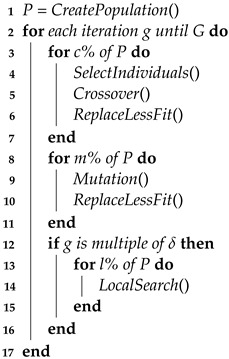



Since randomness is used in various steps throughout the algorithm, the probability of formulating solutions that are invalid in the domain of this problem increases. Therefore, three cases of solution invalidity were identified:individuals that contain repeated genes;individuals that represent solutions that locate the sensors within a range of μ centimeters from each other, referred to as the guard area, in order to take into consideration the physical size of the sensors;configurations that lead to an FIM with singular matrix properties, avoiding using invalid eigenvalues, which occurs, for instance, when the set of sensors are coplanar.

Whenever an algorithm’s routine generates one of the aforementioned cases, the invalid individual is discarded and a new one is created following the same pseudo-random process.

The algorithm converges to an improved feasible solution, with a convergence speed and behaviour that depends on the input parameters of the algorithm. However, as in general heuristic search methods, optimality cannot be guaranteed.

### 3.1. Fitness Function

The multi-objective fitness function for this problem requires a trade-off between the single sensor and the sensor configuration optimization objectives previously described. On the one hand, the improvement of the single sensor evaluation leads to a decrease of the incidence angles $\phi_{k}$ and consequently the penalization level $\rho_{k}$. On the other hand, the sensor configuration evaluation tends to benefit from having larger baselines between sensors to reduce localization uncertainty. Therefore, weights are added to the fitness function in order to balance these criteria.

Firstly, since the penalization level $\rho_{k}$ is mapped to each $\eta_{p_{k}}$ along the deployment surface in relation to a specific beacon position, a global ρ is formulated for each sensor configuration as the average $\rho_{k}$ of the sensor positions that it integrates, $\rho =\sum_{i=1}^{S}\rho_{k_{i}}/S$. Recalling the approaches presented in Section 2, when a double improvement was considered, a map of $\rho_{k}$ was created for each beacon position considered. Therefore, in this case the $\rho_{k}$s are initially computed as the average values obtained in the penalization level maps for both beacon positions, having the global ρ once more as the average of the sensor positions included in the configuration. Regarding the fitness function, in an irregular shaped body the incidence angle along neighbouring positions can vary very rapidly, therefore the localization uncertainty may be very penalized if the same weight is attributed to both optimization criteria. In order to balance these criteria, a rank system was created that attributes the same weight to sufficiently similar ρ values, in order to decrease the sensibility of the fitness function to small ρ variations, since these will not significantly affect the signal quality. Therefore, the ρ values are divided into *X* clusters, and a rank r=[1,…,X] is attributed to each ρ value in ascending order.

Secondly, the variable localization requirements of some applications require the definition of specific constraints for localization uncertainty depending on the reference beacon position. In case of a single improvement, the goal is to optimize localization for a single fixed relative position of the deployment surface in relation to the beacon, thus the fitness function will aim at minimizing the localization uncertainty, $\Lambda_{m}$, in regards to that beacon position. If the application scenario expects a double improvement, where the deployment surface follows a trajectory in proximity to the beacon, two relative beacon positions are selected as reference, thus the optimization process aims at minimizing the uncertainty $\Lambda_{m}$ for short range, when accuracy is more critical, and bound uncertainty $\Lambda_{b}$ for medium/long range, when localization requirements are typically more relaxed. In order to define the tolerable localization uncertainty for medium/long range, a threshold γ is defined according to the specific application. Based on this threshold, a binary function $b_{\Lambda_{b}}$ is generated to classify the solutions according to their feasibility
$$b_{\Lambda_{b}}=\left\{\begin{array}{cc} 0, & \Lambda_{b}\leq \gamma \\ M, & otherwise,\;\;M\;large \end{array}\right..$$

Accordingly, in case of a feasible solution, the uncertainty value $\Lambda_{b}$ is below the threshold γ and $b_{\Lambda_{b}}$ does not change the fitness value. Otherwise, in case of an infeasible solution, $b_{\Lambda_{b}}$ assumes the large number *M*, damaging irreversibly the value of the fitness function for that solution.

Overall, the multi-objective fitness function, F, aims at minimizing the following expression
(10)$$\underset{F}{\arg\min}F=\Lambda_{m}+\left(\Lambda_{m}+b_{\Lambda_{b}}\right)\;r.$$

The fitness function minimizes the localization uncertainty, $\Lambda_{m}$, while defining feasibility of the solution through $b_{\Lambda_{b}}$ and weighting it with the incidence angle ranking *r*. It should be noted that for optimization regarding a single position, $b_{\Lambda_{b}}$ is always equal to zero, meaning that there are no unfeasible solutions based on range uncertainty.

### 3.2. Local Search

The implemented LS routine is based on an iterative first improvement neighbourhood search [37], in which an incumbent solution is substituted by the first solution found in the iterative process that is fitter than the incumbent. Recalling that an individual corresponds to a candidate solution composed by a group of *S* sensors, referred to as genes, it considers a neighbour any individual from the population that only differs from another individual by one gene. Additionally, this gene’s $\eta_{p_{k}}$ is within the LS area, i.e., distanced from the original gene a maximum azimuth range of $\alpha^{{}^{\circ}}$ and a maximum height range of β cm from the original gene.

Algorithm 2 describes the LS routine. Recalling the presented MA, the LS takes place at the end of the MA loop whenever the current iteration *g* of the algorithm is a multiple of δ. For each of the l% fittest individuals of the current population, serving as initial incumbent solutions, an intensive spatial search is performed by substituting each of its genes by neighbours and analysing the resulting fitness.
**Algorithm 2:** Neighbourhood based Local Search**Result:** Local optimized sensor configuration
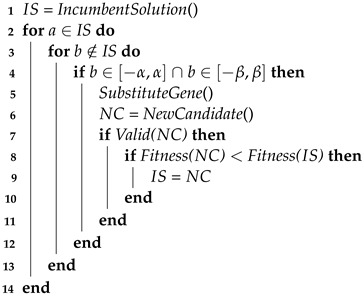


Therefore, for each gene *a* selected randomly from the incumbent individual and for each gene *b* not present in the individual, if gene *b* is within the LS area, then it substitutes *a*. If the new candidate individual is valid, its fitness is calculated and compared with the fitness of the best individual at the moment. In case the new fitness is worse, the candidate is dropped and the search moves to the next iteration, otherwise, as a first improvement algorithm, the new candidate becomes the best solution, from which the search proceeds. The LS finishes when no fitness improvement is detected after searching all genes of the individual that currently holds the best fitness. Figure 8 illustrates an example of LS in an incumbent solution composed by $S_{1}$, $S_{2}$, $S_{3}$ and $S_{4}$, where $S_{1}$ is being tested for replacement by other candidate positions within the LS area to improve fitness, taking into consideration a sensor’s guard areas.

## 4. Simulations and Results

The proposed multi-objective optimization algorithm is suitable for scenarios where it is critical to achieve maximum localization accuracy for a single relative position between the deployment structure and an acoustic beacon, such as docking [38] and “rendezvous” maneuvers in cooperative navigation [39]. Additionally, it also accommodates applications in which the localization requirements are variable with range, for instance in approaching manoeuvres, such as homing and docking [31] or search and recovery. Therefore, this section is dedicated to presenting the potential of the proposed algorithm in a very relevant and increasingly growing application scenario: homing and docking missions of AUVs.

### 4.1. Fundamentals

Underwater docking systems may be used during AUVs missions to recharge batteries and transfer data. These are usually aided with one or more acoustic beacons that serve as localization reference and the AUVs commonly rely on an inverted USBL system [2,31,38,40,41] to estimate the relative localization and approach the station accordingly to a predefined navigation strategy. Although several works add vision-based techniques to improve short range performance [2], these are very dependent on varying environment conditions, as water turbidity, to which acoustic-based techniques are insensible. Therefore, it is relevant to decrease localization uncertainty both during homing and docking. In this regard, a double improvement approach is required and the employed homing and docking strategies dictate the selection of the most relevant beacon’ relative positions for sensor configuration optimization.

In homing, the AUV navigates towards the target station equipped with an acoustic beacon starting from hundreds to thousands of meters of distance, using acoustic signals as reference. The AUV typically points its nose directly at the station; however, it also may navigate with some inclination angle, not directly pointing at the station, depending on the context and navigation approach [42]; for instance if the station is placed directly below the AUV it may not be capable of navigating fully vertically. At this stage, localization requirements are usually more relaxed since the AUV can correct its path in relation to the current localization error; however, the error needs to be bounded to ensure that the path converges to the target [31,43]. Therefore, it is relevant to select a sensor configuration that ensures bounded localization uncertainty for this scenario, where the beacon is directly aligned with the nose of the AUV or, alternatively, at an expected angle depending on the mission context, as illustrated in Figure 9.

In close proximity to the station of a few meters, accurate relative position is essential to avoid collision when performing manoeuvres, allowing correct landing and effective attachment to the system for battery recharge, data transfer, among others. Docking strategies vary depending on the type of AUV and docking station, that influences the selection of relevant relative positions for sensor configuration optimization. Two of the most typical scenarios will be considered:*Flying AUVs* [40]: The docking station is usually shaped as a funnel or cone where the AUV is expected to enter. When in close proximity, the AUV must be aligned with the entrance to correctly enter the station. Therefore, the aim is to minimize localization uncertainty for a relative position of the AUV directly at the entrance of the station, thus a beacon position aligned with the nose of the AUV, as illustrated in Figure 9a.*Hovering AUVs* [44]: The docking station usually has an entrance above where the AUV is expected to anchor. The AUV approaches the station and starts by positioning itself directly above it, distanced at a few meters. Then it continues to approximate vertically to the station’s surface until reaching the docking platform. In this approach, the goal is to minimize localization uncertainty for a relative position of the beacon aligned with the underside of the AUV, as illustrated in Figure 9b.

In order to combine the homing and docking localization requirements presented above, the multi-objective optimization algorithm should select a sensor configuration that minimizes localization uncertainty to the most relevant docking relative position between the station and the AUV, bounds localization uncertainty based on a defined threshold to ensure homing and improves incidence angle of the acoustic signals.

### 4.2. Simulation and Results

In order to evaluate the implemented algorithm, a set of simulations were formulated using inverted USBL sensor configurations in both flying and hovering type AUVs for homing and docking scenarios. Four single improvement and four double improvement cases will be presented, followed by an additional adaptability analysis adding forbidden areas for sensor placement in the AUVs body.

An initial parameter tuning was performed to find the best combination of the memetic algorithm parameters. The values used in the remainder of this section are defined in Table 1.

Some of the main characteristics provided by the selected memetic algorithm parameters are increased gene variability in the population, variation of the population, intensification of the local search in adequate stages of the evaluation and convergence of the fitness function that leads to a higher chance of reaching the optimal solution [28]. The number of sensors *S* per solution is 4, since this is the minimum to avoid ambiguity in 3D localization based on time measurements. The threshold γ is expressed by the ratio between the maximum acceptable localization uncertainty for medium/long range, $\Lambda_{b}$, and the distance between the reference beacon and the closest point of the AUVs body, *d*. This value is influenced by the ability of the AUVs navigation and control mechanisms to converge to the reference beacon; thus, it is dependent on the application and context. In this case, a 15% threshold is considered when the distance *d* is 500 m, that consists on an uncertainty of 75 m and a deviation of approximately 8.5° which is assumed to ensure convergence.

The simplified torpedo shaped AUV considered in this section has an 1.6 m of length and a diameter of 0.4 m. The considered reference frame is fixed in the AUVs body, having the vehicle aligned with the *z* axis, the edge of the nose positioned in [001.6] m, in Cartesian coordinates, and the opposite side set on plane xy. Throughout this section, the AUV is assumed to be fixed in the reference frame and the reference beacon positions will be updated to define different relative positions to the AUV.

The performed simulations intend to demonstrate the sensor placement optimization in homing and docking of both flying and hovering AUV, as illustrated in Figure 9. Four distinct reference beacon positions were chosen based on relevant orientations between the AUV and the beacon during homing and docking, defined as follows:R${}_{1}$ represents a beacon distanced 0.5 m from the AUVs nose, to simulate a critical relative position of docking for a flying AUV;R${}_{2}$ characterizes a beacon distanced 500 m from the AUVs nose, representing a homing mission of an AUV navigating with the nose heading to the station, that can occur both with flying and hovering AUVs;R${}_{3}$ represents a beacon distanced 0.5 m from the AUVs side, to simulate the final stage of a docking approach for a hovering AUV;R${}_{4}$ illustrates a beacon distanced 500 m from the AUVs side, representing a homing approach in which the AUV does not point the nose directly at the reference and navigates at an approximated angle of 50° with the direct path reference.

Assuming these reference beacon positions, an initial set of simulations were performed using a single improvement approach, aiming at minimizing localization uncertainty $\Lambda_{m}$ while reducing the penalization level ρ, due to the contribution of the incidence angle.

Table 2 summarizes the simulation’s details, recalling that the inputs are *d*, the distance between the reference beacon and the closest point of the AUVs body, and $\eta_{b}$, the Cartesian coordinates of the reference beacon. For each reference, the optimization algorithm outputs a sensor configuration solution, with sensor positions $\eta_{s_{1}}$, $\eta_{s_{2}}$, $\eta_{s_{3}}$ and $\eta_{s_{4}}$, that results in a localization uncertainty of $\Lambda_{m}$ and the total penalization level ρ.

Figure 10 illustrates the resulting sensor configurations for the several simulated cases. For each reference beacon, the penalization level is mapped with colors in the AUVs surface. The color ranges from yellow, representing of the most penalized areas with higher incidence angle, to blue, illustrating the most favourable areas for sensor placement with lower incidence angle. Beacon positions are represented by the black segmented shapes. The red dots represent the positions $\eta_{s_{1}}$, $\eta_{s_{2}}$, $\eta_{s_{3}}$ and $\eta_{s_{4}}$ of the optimized sensor configuration.

It can be observed that reference beacon positions that form reduced low penalization areas in the AUV body, particularly R${}_{1}$ and R${}_{2}$, tend to concentrate the sensors in a small area, in order to contain the incidence angle and consequently penalization level. This results in a reduced baseline between sensors that leads to increased localization uncertainty. Additionally, the observed penalization levels are similar for all cases, maintain the average incidence angle between 35° and 37°. For R${}_{1}$, the beacon position is aligned with the AUVs nose and, since incidence angle varies more rapidly for short range positions, R${}_{1}$ represents the case with larger localization uncertainty. In practice, since the larger uncertainty eigenvector is along the direction of the reference beacon, R${}_{2}$ has a slightly larger low penalization area due to an increased range between the AUV and the beacon, therefore a decrease in uncertainty is observed. In R${}_{3}$ the beacon is positioned on the side of the AUV, forming a larger low penalization area and the surface is more flat allowing more distance between sensors without penalizing incidence angle. Lastly, in R${}_{4}$ the beacon reaches a much larger area with LOS, therefore the configuration sensors are spread out through the surface, reaching lower uncertainty while compensating incidence angle by averaging the penalization level of all sensor positions.

It should be noted that in the case that a simulated combination of surface shape and reference beacon position cannot ensure the predefined threshold γ for $\Lambda_{b}$, no feasible solution is returned by the algorithm.

Having the previous single improvement analysis as foundation, double improvement scenarios of homing and docking were formulated by combining R${}_{1}$, R${}_{2}$, R${}_{3}$ and R${}_{4}$. The goal in this approach was to minimize localization uncertainty $\Lambda_{m}$ for short range beacon positions and bound localization uncertainty $\Lambda_{b}$ for medium/long range beacon positions, while reducing the penalization level ρ.

Table 3 summarizes the obtained solutions. For each double improvement case, the combination of reference beacons R${}_{c}$ and R${}_{d}$ is referred to as R${}_{cd}$. The optimization algorithm outputs a sensor configuration, formed by sensor positions $\eta_{s_{1}}$, $\eta_{s_{2}}$, $\eta_{s_{3}}$ and $\eta_{s_{4}}$, that results in a localization uncertainty for the short range position $\Lambda_{m}$, a localization uncertainty for the medium/long range position $\Lambda_{b}$ and the total penalization level ρ.

Figure 11 presents the resulting sensor configurations for each case, followed by the illustrative representation of the corresponding homing and docking scenario. The penalization color map represented in the AUVs body corresponds to the averaged individual maps for each reference beacon, resulting in an increase of the general penalization along the surface.

Overall, the following assessments can be made based on the observed results:In R${}_{12}$, the penalization maps, generated by R${}_{1}$ and R${}_{2}$, were very similar and the low penalization area was very constrained. Therefore, an improvement in $\Lambda_{m}$ in relation to R${}_{1}$ was observed; due to the slight enlargement of the low penalization area, the $\Lambda_{b}$ was approximately the same and under the defined threshold and ρ increased, corresponding to an approximated average incidence angle of 41°;In R${}_{14}$, the combination of a very constrained penalization map from R${}_{1}$ and a more relaxed one from R${}_{4}$, resulted in an attenuation of the less prominent low penalization areas. Therefore, $\Lambda_{m}$ was improved in comparison with R${}_{1}$; due to the larger low penalization area, $\Lambda_{b}$ remained below the defined threshold but worsened in relation to R${}_{4}$, due to the spatial concentration of the low penalization area, and ρ increased slightly, producing an average incidence angle of approximately 39°;In R${}_{32}$, the combined penalization maps had concentrated low penalization areas in distinct spatial locations of the surface, resulting in a global degradation of the overall penalization level for the Therefore, $\Lambda_{m}$ deteriorated slightly in relation to R${}_{3}$, $\Lambda_{b}$ complied with the defined threshold and improved in relation to R${}_{2}$, due to the low penalization area increment. The value of ρ increased significantly, to an average incidence angle of approximately 51°, since the reference beacon positions had divergent incidence orientations;In R${}_{34}$, the overlaid maps resulted in a dispersed area of low penalization that allowed increased sensor baselines while containing the penalization level. Therefore, $\Lambda_{m}$ improved slightly in relation to R${}_{3}$, $\Lambda_{b}$ deteriorated in relation to R${}_{4}$ but remained under the defined threshold. The value of ρ increased to an average incidence angle of approximately 49°.

After analyzing the method’s performance in homing and docking scenarios, a last scenario was simulated to test the adaptability of the algorithm. Therefore, forbidden areas were added to the AUV surface by attributing maximum penalization to selected segments, as a way to constraint the sensor placement. In order to simulate this, the MARES [45] AUV was used as a model, represented in Figure 12. The selected forbidden areas for sensor placement in this case was the surface around the thrusters, since these were a significant source of noise, and the sonar, due to spatial incompatibility.

The scenario of reference R${}_{34}$ was used to test this feature and the results of the simulation are presented in Table 4. A degradation of $\Lambda_{m}$ in relation to R${}_{34}$ was observed, as expected since the sensor placement was more constrained, while $\Lambda_{b}$ slightly improved since an additional sensor moved to the most relevant area for the homing reference, or the medium/long range. The value of ρ decreased slightly, due to the sensor placement alteration, to an average incidence angle of approximately 47°.

Although the used AUV model is considerably simple, this algorithm is also applicable in more complex AUV shapes, for example integrating wings or remotely operated underwater vehicles (ROVs). In these cases, the spatial complexity of the surface can be exploited to improve localization with a reduced incidence angle, and avoiding placing sensors in fragile, noisy or other constrained areas.

## 5. Conclusions

The present work proposes a multi-objective memetic algorithm for acoustic sensor placement in an irregular and constrained 3D surface to improve localization of an acoustic beacon. The implemented algorithm combines the expected quality of the measurements, affected by the incidence angle of acoustic signals arriving at each sensor on the deployment body, with the information acquired from the localization uncertainty associated with the spatial geometry of a sensor configuration. Additionally, the sensor configuration solutions can follow a single improvement approach, that aims at minimizing the localization uncertainty for a single acoustic beacon position, or a double improvement approach, that aims at simultaneously minimizing and bounding uncertainty based on the range between the deployment body and the reference beacon positions.

A case study was addressed considering homing and docking scenarios with flying and hovering AUVs. Considering these scenarios, there are many unknown details regarding the application context, for instance navigation and control strategies, that influence the suitability of the results in a real application. However, by optimizing the sensor placement, having taken into consideration both the quality and the information acquired from the measurements, we believe that the resulting sensor configurations will improve localization in real field applications. Lastly, adding forbidden areas allows real and complex vehicles to be modelled as simplified structures, improving computational performance and widening the applications spectrum for this sensor placement optimization methodology.

Future work includes exploiting alternative ways of integrating the acoustic sensors’ sensitivity response in the fitness function. In the current formulation, the orientation of the individual sensors with respect to the acoustic source was considered as a penalization in the fitness function, as the average of the incidence angles of the acoustic wave to all the acoustic sensors. This penalization aims to contemplate a non-omnidirectional sensitivity reception diagram that is expected for acoustic receivers placed on a non-planar surface, or even partially buried in the structure wall. As acoustic localization procedures rely on the determination of the time of arrival of acoustic signals, an alternative, and possibly better, way to include this metric in the fitness function is to translate the incidence angle of the arriving acoustic wave of each sensor in order to calculate an uncertainty of the time of arrival for that sensor.

The scenarios considered in this work allow the position of a set of acoustic receivers along a given surface to be optimized. Although this approach might be of interest when setting up a set of acoustic sensors for localization on the body of a vehicle, the fixed position of such sensors would naturally compromise the operation under different scenarios. A variant of this problem considers a fixed, pre-determined set of *M* discrete locations for placing the *S* acoustic sensors (S<<M) required to implement the localization processes, and optimize the set of *S* sensors in use to improve the localization accuracy. With the capability of building arrays of custom acoustic sensors attached to a vehicle’s surface, an adaptive localization system may choose, in real-time, the best set of *S* sensors, out of the larger set *M*, that optimize the localization process for the current relative position of the reference transmitting beacon. Another related and more generalized optimization problem is to optimize the *M* locations where to place the acoustic sensors, from which only *S* will be dynamically selected for the localization process.

These problems are the focus of the ongoing research aiming to build adaptive, and thus more flexible, acoustic localization systems for the operation of underwater vehicles.

## Figures and Tables

**Figure 1 sensors-23-06316-f001:**
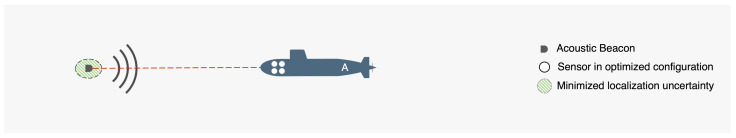
Single improvement: an illustrative deployment body A carries an optimized sensor configuration that provides minimized localization uncertainty around the acoustic beacon (green region).

**Figure 2 sensors-23-06316-f002:**
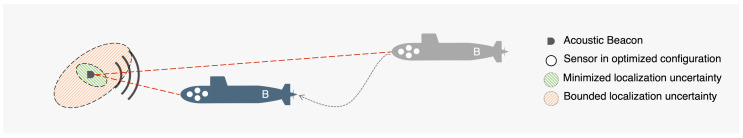
Double improvement: an illustrative deployment body B carries an optimized sensor configuration that simultaneously minimizes localization uncertainty for a short range position (green region) while ensuring bounded localization uncertainty for a medium/long range position (orange region).

**Figure 3 sensors-23-06316-f003:**
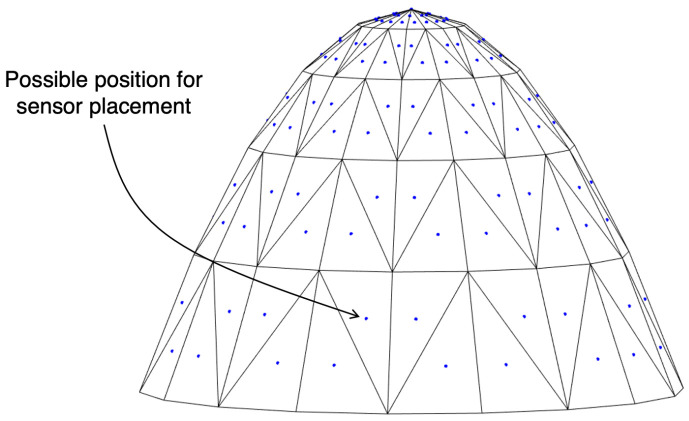
Example of a continuous surface generated from Delaunay triangulation, where the centroids of each triangle represent possible positions for sensor placement.

**Figure 4 sensors-23-06316-f004:**
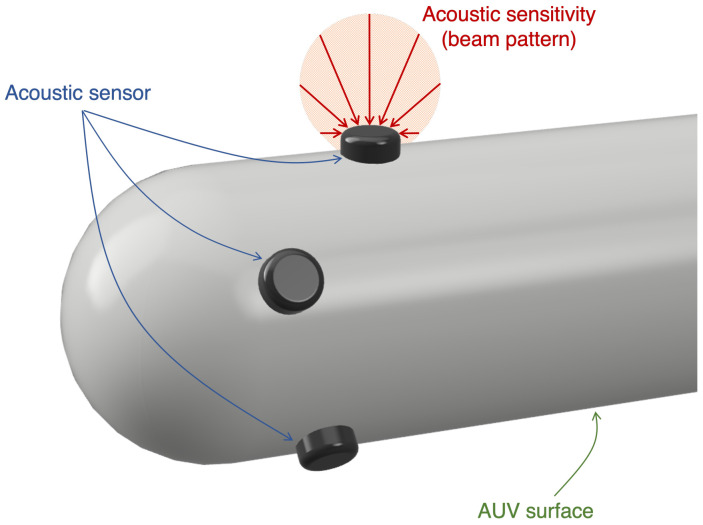
Illustration of the acoustic sensors placed on the non-planar 3D surface of an AUV. The sensor’s acoustic sensitivity response is represented by a beam pattern, where the arrows describe different incidence angles of the received signal and their size corresponds to the attenuation.

**Figure 5 sensors-23-06316-f005:**
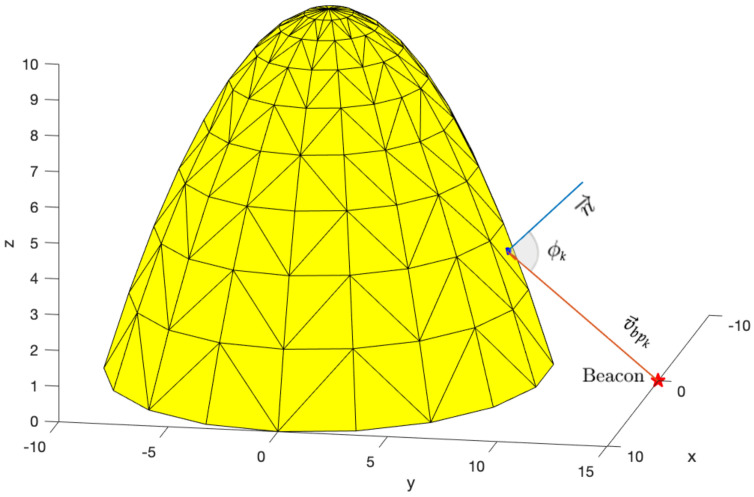
Example of a parabolic shape formed by Delaunay triangulation and representation of the incidence angle $\phi_{k}$ between $\vec{n}$ and ${\vec{v}}_{bp_{k}}$ for an arbitrary candidate position for sensor placement, $\eta_{p_{k}}$.

**Figure 6 sensors-23-06316-f006:**
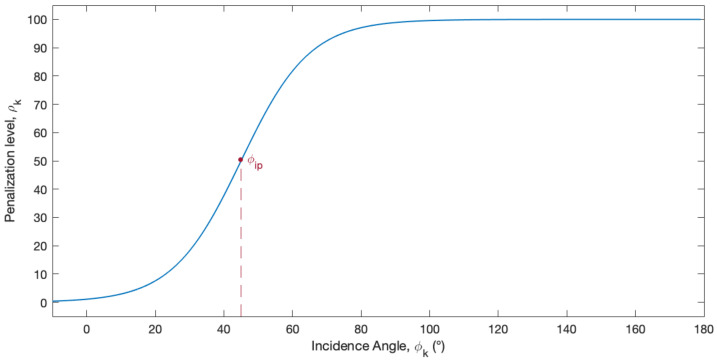
Model of the acoustic sensors’ beam pattern as a sigmoid function that describes the penalization level $\rho_{k}$ according to the incidence angle $\phi_{k}$ of acoustic waves arriving at the sensor. The used inflexion point value for this representation is $\phi_{ip}$ = 45°.

**Figure 7 sensors-23-06316-f007:**
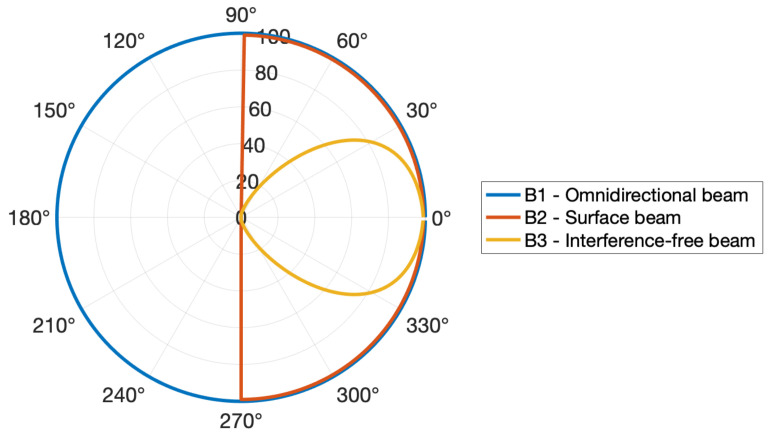
Illustrative beam patterns of acoustic sensors, where the angle represents the incidence angle, $\phi_{k}$, and the radius corresponds to the penalization level, $\rho_{k}$. B1 corresponds to an omnidirectional beam pattern, B2 is the beam pattern for an omnidirectional sensor placed on a surface and B3 represents the directional beam pattern of the sensors in the proposed methodology.

**Figure 8 sensors-23-06316-f008:**
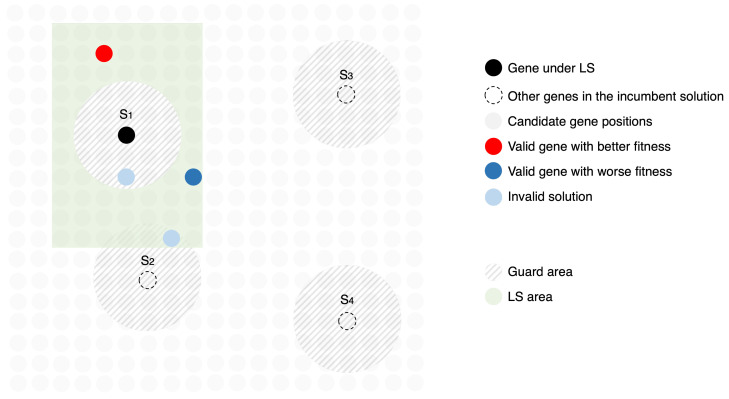
Illustrative example of local search of an incumbent solution composed by four sensors $S_{1}$, $S_{2}$, $S_{3}$ and $S_{4}$, referred to as genes. In this phase, gene $S_{1}$ is being substituted by candidate genes within the LS area, from which examples of possible outcomes are highlighted: (a) in red, a valid gene that leads to a better fitness solution than the incumbent, therefore will substitute gene $S_{1}$ in the following incumbent solution; (b) in dark blue, a valid gene that leads to a worse fitness solution than the incumbent, therefore is discarded; (c) in light blue, invalid solutions that invade sensor’s guard areas.

**Figure 9 sensors-23-06316-f009:**
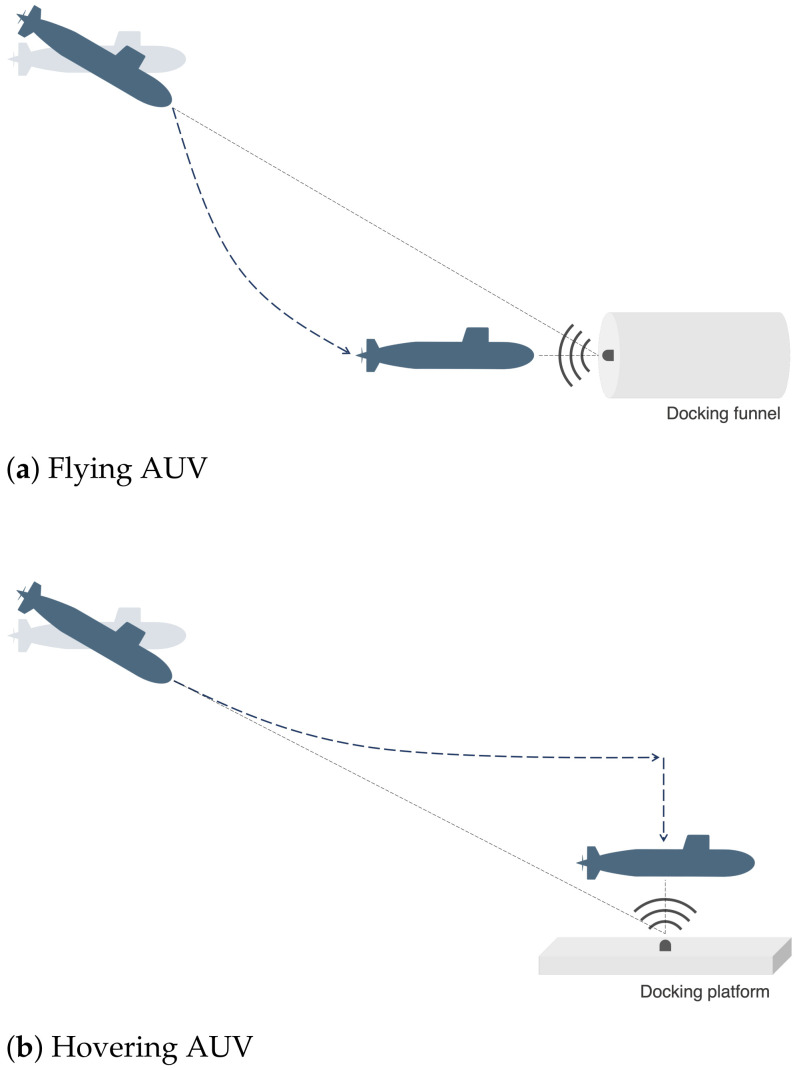
Homing and docking strategies.

**Figure 10 sensors-23-06316-f010:**
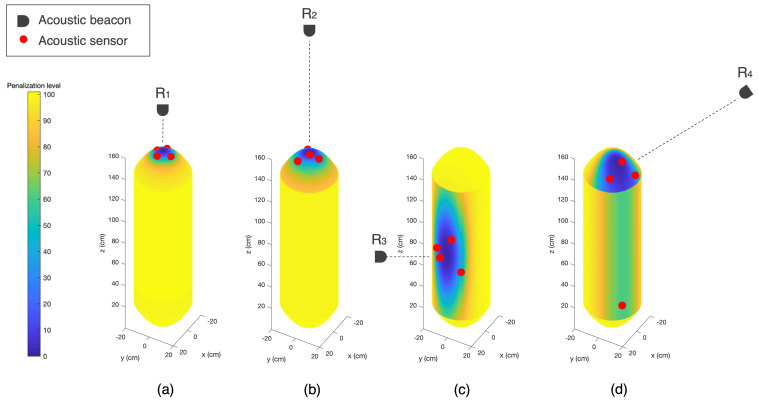
Single improvement scenarios: (**a**) docking of flying AUV, using reference beacon R${}_{1}$; (**b**) docking of hovering AUV, using reference beacon R${}_{2}$; (**c**) homing with nose directly pointed at the station, using reference beacon R${}_{3}$; (**d**) homing with an inclination angle to the station, using reference beacon R${}_{4}$.

**Figure 11 sensors-23-06316-f011:**
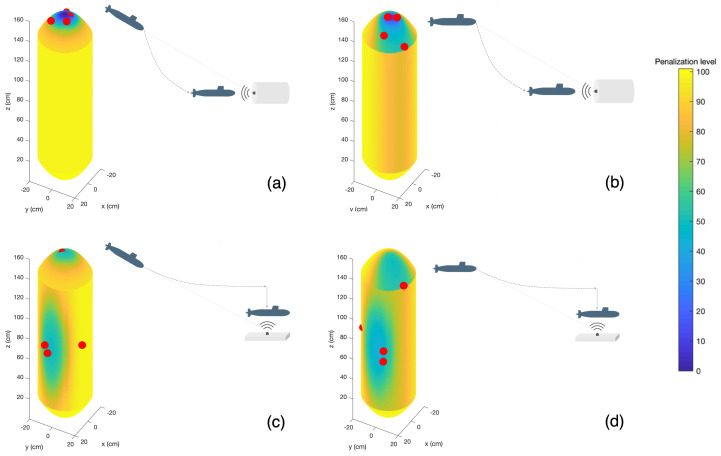
Double improvement scenarios: (**a**) R${}_{12}$ combines reference R${}_{1}$ for docking and R${}_{2}$ for homing, assuming a flying AUV; (**b**) R${}_{14}$ combines reference R${}_{1}$ for docking and R${}_{4}$ for homing, assuming a flying AUV; (**c**) R${}_{32}$ combines reference R${}_{3}$ for docking and R${}_{2}$ for homing, assuming a hovering AUV; (**d**) R${}_{34}$ combines reference R${}_{3}$ for docking and R${}_{4}$ for homing, assuming a hovering AUV.

**Figure 12 sensors-23-06316-f012:**
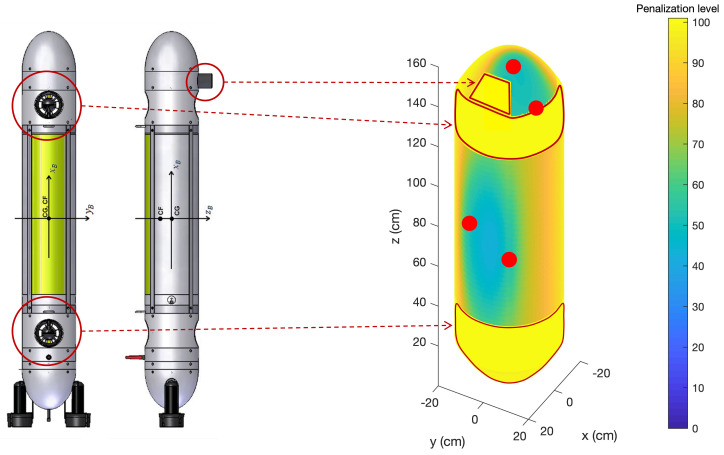
MARES model [45] (**left**) and simulation representation of the AUV with added forbidden areas and the optimized sensor configuration given by the algorithm (**right**).

**Table 1 sensors-23-06316-t001:** Summary of the input and output variables of the memetic algorithm.

Input		Value
*S*	Number of sensors per configuration	4
*N*	Population size	1000
*G*	Number of generations	500
*c*	Fraction of population to which crossover is applied	40%
*m*	Fraction of population to which mutation is applied	5%
*l*	Fraction of population to which local search is applied	20%
μ	Radius of guard area	0.06 m
δ	LS execution frequency (no. of iterations)	50
α	Azimuth of LS area	20°
β	Height of LS area	0.1 m
γ	Threshold for medium/long range uncertainty	15%
**Output**		
F	Fitness value	
ρ	Penalization level	
ε	Localization uncertainty	

**Table 2 sensors-23-06316-t002:** Simulation results for single improvement scenarios.

Ref	Reference Beacon Position		Fitness Function		
	*d*(m)	$\eta_{b}$(m)	$\Lambda_{m}$(m)	$\Lambda_{m}$/*d*(%)	ρ
R${}_{1}$	0.5	[0 0 2.1]	0.095	19	29.9
R${}_{2}$	500	[0 0 501.6]	70.74	14.1	30.0
R${}_{3}$	0.5	[0.7 0 0.8]	0.044	8.8	26.7
R${}_{4}$	500	[206 250 380]	26.16	5.2	28.4
**Ref**	**Final Sensor Coordinates**				
	$\eta_{s_{1}}$ **(** $10^{-2}$ **m)**	$\eta_{s_{2}}$ **(** $10^{-2}$ **m)**	$\eta_{s_{3}}$ **(** $10^{-2}$ **m)**	$\eta_{s_{4}}$ **(** $10^{-2}$ **m)**	
R${}_{1}$	[5.3 −1.6 158.4]	[−0.6 −5.1 158.7]	[−6.4 −0.4 158]	[0.4 6.3 158]	
R${}_{2}$	[7.7 −5.1 155.7]	[1.3 1.4 159.8]	[−0.4 7.6 157.1]	[−7.2 −5.4 155.8]	
R${}_{3}$	[17.3 10 64.1]	[18.8 −6.8 71.4]	[17.3 −10 78.8]	[19.7 3.5 93.5]	
R${}_{4}$	[14.1 6.1 148.2]	[−2.1 17.5 144.4]	[12.9 15.3 32.2]	[2.0 8.6 156.1]	

**Table 3 sensors-23-06316-t003:** Simulation results for double improvement scenarios: a flying AUV in R${}_{12}$ and R${}_{14}$ and a hovering AUV in R${}_{32}$ and R${}_{34}$.

Ref	Fitness Function				
	$\Lambda_{m}$(m)	$\Lambda_{m}$/*d*(%)	$\Lambda_{b}$(m)	$\Lambda_{b}$/*d*(%)	ρ
R${}_{12}$	0.075	15	69.10	13.8	39.96
R${}_{14}$	0.083	16.6	68.28	13.6	35.54
R${}_{32}$	0.047	9.4	53.69	10.7	64.55
R${}_{34}$	0.042	8.4	50.79	10.1	59.82
**Ref**	**Final Sensor Coordinates**				
	$\eta_{s_{1}}$ **(** $10^{-2}$ **m)**	$\eta_{s_{2}}$ **(** $10^{-2}$ **m)**	$\eta_{s_{3}}$ **(** $10^{-2}$ **m)**	$\eta_{s_{4}}$ **(** $10^{-2}$ **m)**	
R${}_{12}$	[4.2 4.0 158.3]	[−2.1 3.2 159.3]	[−8.8 −3.8 155.3]	[4.8 −9.6 154.2]	
R${}_{14}$	[13.3 3.2 150.6]	[−1.6 5.3 158.4]	[2.3 0.1 159.7]	[7.6 17.7 141.3]	
R${}_{32}$	[18.8 −6.8 78.8]	[−0.2 −2.7 159.6]	[19.7 −3.5 73.9]	[6.8 18.8 81.2]	
R${}_{34}$	[7.8 18.1 140.5]	[18.8 6.8 69.0]	[18.8 6.8 78.8]	[12.9 −15.3 88.6]	

**Table 4 sensors-23-06316-t004:** Simulation results for adding forbidden areas to MARES AUV model in reference scenario R${}_{34}$.

Ref	Fitness Function				
	$\Lambda_{m}$(m)	$\Lambda_{m}$/*d*(%)	$\Lambda_{b}$(m)	$\Lambda_{b}$/*d*(%)	ρ
R${}_{34}$	0.057	11	47.28	9	54.8
**Ref**	**Final Sensor Coordinates**				
	$\eta_{s_{1}}$ **(** $10^{-2}$ **m)**	$\eta_{s_{2}}$ **(** $10^{-2}$ **m)**	$\eta_{s_{3}}$ **(** $10^{-2}$ **m)**	$\eta_{s_{4}}$ **(** $10^{-2}$ **m)**	
R${}_{34}$	[4.2 4.0 158.3]	[18.8 −6.8 88.6]	[18.8 −6.8 88.6]	[7.4 17.1 142.6]	

## Data Availability

Not applicable.

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
