# Peer review of "Sensor Placement in an Irregular 3D Surface for Improving Localization Accuracy Using a Multi-Objective Memetic Algorithm"

_sensors, 2023, doi:10.3390/s23146316_

Round 1

Reviewer 1 Report

[Comment 1] Novelty

[Subcomment 1a] The authors must show that memetic algorithm is the state-of-the-art method, while citing the reference, especially for the discussed problem.

[Subcomment 1b] The authors must compare their proposed memetic algorithm with other methods.

[Subcomment 1c] Did the authors used any existing memetic algorithm 100% without any modification? Please clearly cite the reference used for the algorithm and the changes made by the authors, if exist.

[Comment 2] Proposed methods

[Subcomment 2a] I believe that the authors have not mentioned the number of beacons to install yet. Please explain about it, and how the beacons should be located relatively from each other.

[Subcomment 2b] The authors seem to optimize the placement of the beacons. The authors should first define the optimization problem, by clearly mentioning the (1) objectives (minimization or maximization), (2) decision variables, and (3) constraints.

[Comment 3] Writing quality and clarity

(lines 243-249) Please illustrate the situation using a figure (including the mentioned triangle), for clarity.

"unbias estimator" should be revised into "unbiased estimator"?

Author Response

Dear Editor,

We appreciate the opportunity to submit a revised draft of our manuscript to MDPI Sensors. We are grateful to the Editor and to the Reviewers for their effort and for the insightful comments they provided to improve our paper in this review round. We have carefully considered the raised matters and applied our best effort to satisfactorily address them.

We kindly ask you to consider the revised manuscript attached to this letter, where the major changes are
highlighted, as well as the detailed answers to all comments raised by the Reviewers.

We sincerely hope that this revised version of the manuscript meets the expectations of the Reviewers and of
the high standards of the MDPI Sensors journal. Please notify us if any additional information is required.

Sincerely,

Paula A. Graça,
on behalf of the authors

Reviewer 2 Report

A problem of optimum acoustic sensor placement in AUV surface is solved with a multi-objective memetic algorithm. The manuscript is clear and well written. However, I would recommend to discuss the following questions as well

1. AUV is considered in a free space, that looks oversimplified. However, it works in a complex underwater environment. The received signal is composed of many arrivals (direct, bottom reflected, water surface reflected, refracted etc.), which have different incidence angles at the sensors. Another obstacle is the presence of currents, internal waves, surface waves, sound speed gradients that can degrade the performance of your sensors even if their placements are perfect in a free space. Please, discuss the limitations of your approach from underwater acoustics point of view.

2. How the localization accuracy was improved using your approach? I mean, in term of cm or degrees, not penalization. What signal-to-noise ratio is needed for this improvement? This information will be very useful for real applications.

3. What type of acoustic signals are used for localization problem? What is the sound frequency? If low frequency is used, the sensor will be allowed to "hear the sound" even at the opposite site of AUV.

4. In many applications, including AUV, localization can be performed using vector sensors. It is enough to have one such sensor to do localization. Please, make some comments on it.

sincerely,

reviewer

Author Response

(The authors gave the same response as above.)

Round 2

Reviewer 1 Report

Thank you for your revisions.